# TNM Tumor Classification from Unstructured Breast Cancer Pathology Reports using LoRA Finetuning of Mistral 7B

## Kyle McCleary[1], James Ghawaly[1], Lucio Miele[2]

[1]Louisiana State University, [2]LSU Health New Orleans
kmccl24@lsu.edu, jghawaly@lsu.edu, lmiele@lsuhsc.edu

## Abstract

Over the past year, large language models have seen an explosion in usage, with researchers and companies rushing to discover new applications. This explosion was kick-started by OpenAI, with their release of GPT 3.5 and GPT 4 to the general public. These foundation models have proven extraordinarily capable on a wide range of tasks, but their cost and reliability present problems for more sensitive and/or resource-limited applications. Over the same time-span, however, we have also seen a rush of development in smaller foundation models, such as Mistral's 7B model, as well as in fine-tuning those models for specific tasks.

In this paper, we explore the application of Low-Rank Adaptation (LoRA) fine-tuning of small language models for performing TNM (Tumor, Lymph Node, Metastasis) staging on unstructured pathology reports for triple negative breast cancer cases. We also attempt to develop a more generalized approach, so that our work can be applied to other NLP tasks within the medical field.

We found that performing TNM staging with reliable accuracy is possible for a small foundational model through fine-tuning, allowing fast and reliable automation of critical language processing tasks within medicine.

## Introduction

The field of natural language processing (NLP) has witnessed remarkable advancements in the past year and a half, fueled by the advent of large language models (LLMs) like GPT-3.5 (OpenAI 2022) and GPT-4 (OpenAI et al. 2023). These models have demonstrated impressive capabilities across a diverse range of tasks, ushering in an explosion of research and industry interest in LLMs. However, the widespread adoption of such models comes with challenges, including cost considerations and reliability concerns, especially in sensitive or resource-constrained applications such as medicine.

In parallel with the development of large-scale models, there has been a growing interest in leveraging smaller foundational models and more efficient fine-tuning strategies to address specific task requirements. Mistral's 7B model (Jiang et al. 2023), a representative of this trend, serves as the focal point of our investigation. This shift towards

smaller models offers potential solutions to the challenges posed by their larger counterparts, making them more accessible for adaptation to specialized tasks. These models allow much cheaper solutions, direct control over the model, as well as the option to self-host without reliance on a third party.

In this paper, we delve into the application of LoRA (Hu et al. 2021) fine-tuning of small foundational models for domain-specific tasks. Our primary focus is on TNM malignant tumor classification using unstructured pathology reports. Specifically, we concentrate on triple-negative breast cancer cases, aiming to automate the TNM staging process through a fine-tuned version of Mistral 7B Instruct.

The motivation behind our work is twofold: first, to demonstrate the effectiveness of LoRA finetuning for small language models in medical NLP tasks, and second, to contribute towards a more generalized approach that can be extended to various tasks within the medical domain. In addition to our results on TNM classification, we also demonstrate that fine-tuning a model on only 100 samples can significantly improve performance on this task. This finding may enable physicians to automate similar tasks in a short period of time by manually creating or procuring small datasets.

## Data Synthesis and Validation

We were provided roughly 200 digital reports by the Louisiana Tumor Registry. These reports were provided to us as photo-scanned PDFs with all personal identifiable information redacted. These reports were utilized to develop a process for generating new synthetic reports and data, which was crucial for obtaining enough data to train and validate our model. Given that the original reports did not contain any ground truth labels, subject matter experts on our team manually labeled the data.

The data synthesis process began by converting the provided reports to text using optical character recognition. A script then stripped each report of all information relevant to TNM staging, with the resulting stripped report being used as templates for the following process. Markers were placed in the stripped reports at locations where new statements could reasonably be inserted. A large JSON document of every possible value in the three categories of TNM was then created (More on this in **Language Sampling**),

| Model | T | N | M | Avg | Compounded[*] | Conf[‡] | Training Time | Samples |
|---|---|---|---|---|---|---|---|---|
| Mistral 7B Instruct | 47.8 | 52.0 | 40.3 | 46.7 | 10.0 | 97.1 | N/A | 200 |
| GPT 3.5 Turbo (1106) | 58.4 | 75.4 | 45.3 | 59.7 | 19.9 | 73.4 | N/A | 100 |
| GPT 4 Turbo (1106) | 64.5 | 84.7 | 57.4 | 68.9 | 31.4 | 93.7 | N/A | 50 |
| Instruct Fine-tuned (100@4)[†] | 89.0 | 90.4 | 97.4 | 92.3 | 78.4 | 99.4 | 1 Hr | 200 |
| Instruct Fine-tuned (100@16)[†] | 89.0 | 91.0 | 97.5 | 92.4 | 78.9 | 100.0 | 3 Hrs | 200 |
| Instruct Fine-tuned (800@4)[†] | 87.5 | 99.5 | 98.0 | 95.0 | 85.3 | 99.6 | 8 Hrs | 200 |
| Instruct Fine-tuned (1600@4)[†] | **97.0** | **100** | **99.5** | **98.8** | **96.5** | 99.4 | 17 Hrs | 200 |

Table 1: Performance Metrics of Various Models Across Categories
[*]Compounded indicates the likelihood of getting all three categories correct on a given pass.
[†]Note the notation, 100@16, indicates fine-tuning on 100 samples to 16 epochs
[‡]The likelihood that the model does not yield 'UNKNOWN' on samples with known staging values.

along with definitions of the categories and a selection of example sentences indicating the corresponding category. Finally, a script was then written to randomly generate new sample sentences from the aforementioned JSON document that indicate a specific TNM category. These sentences were then injected into the template reports at randomly selected marker locations. This process is outlined in Figure 1. In the sampling process, there was also a user-defined low probability that sentences indicating a specific classification category were omitted, indicating the response 'UNKNOWN' in order to train the model to communicate a lack of confidence when the information needed to make the decision is unavailable in the report. We also used separate sets of report templates for training and evaluation, so that no template is seen during both.

**Language Sampling**

As stated above, we created a large structured document of all possible classes in the T, N, and M categories. In this document, each class was filled with a definition from existing literature, notes on the category, example vocabulary indicating the category, and sample sentences indicating the category which were representative of language we would find in real reports. The sample sentences field often included around 20 sentences for almost all categories, although some special categories were more limited due to very straightforward class definitions. In some cases, we also included a 'negative' sample sentences field, which included example language indicating that the class was not applicable. These sentence groups, and effectively the document, were split into a 4:1 ratio for training and evaluation, respectively. However, this was done on the condition that there be at least 10 sample sentences in the class, otherwise the full set would be used for both training and validation. Exceptions included a handful of specialized categories which were typically very direct in definition and language, and often had only a handful of ways to express that the category was applicable. For example, the T category of Paget's Disease requires the use of the term 'Paget' in some form. These classes were largely expressed by only a single term, making the task quite straightforward in these cases.

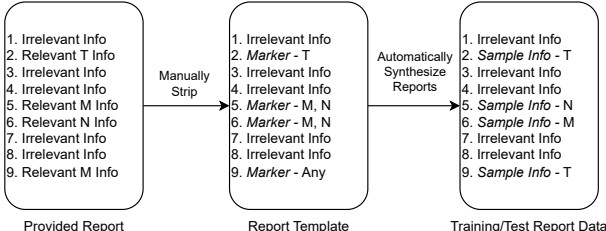

Figure 1: Simplified outline of the report synthesis process.

While the use of sample sentences prevented us from validating on the full scope of natural language, splitting the document into a training and evaluation set ensured that the statements observed during testing were completely different from the ones used for training, with the exception of the few limited categories described previously. Along with the use of separate templates for training and evaluation, we believe that this allowed us to properly test the model's generalized performance, as the training and test data shared almost nothing.

**Training**

During the report synthesis process described in Section 2 and outlined in Figure 1, we formatted our training results into the prompt templates that we would be using during evaluation, along with the expected responses. We then used these reports to finetune Mistral's 7B Instruct model via LoRA. We opted to use the Axolotl library for fine-tuning, and evaluation was performed with several different sample counts.

Our most extensive training utilized 1600 samples and was trained to 4 epochs, which took roughly 17 hours on a single RTX 4090 GPU. The Axolotl configuration file for our training is contained in our source code, which we've published. Fine-tuning was done with a rank of 64 (all other fine-tunes utilized a rank of 32), and training targeted the $Q_{proj}$, $V_{proj}$, $K_{proj}$, $O_{proj}$, $Gate_{proj}$, $Down_{proj}$, and $Up_{proj}$ modules within the original model, accounting for

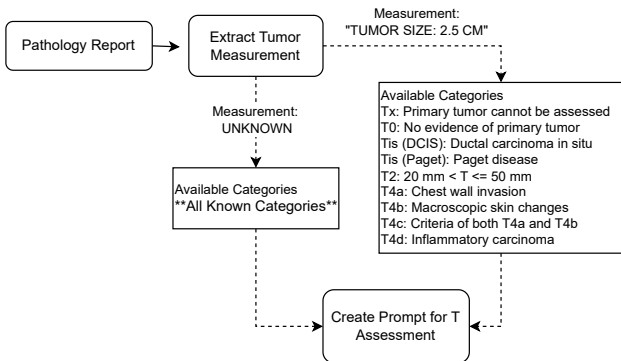

Figure 2: The class filtering performed during T evaluation using measurement information extraction.

roughly 90 million of the model's 7 billion weights.

## Evaluation

Our evaluation consists of four passes to the LLM. First, we attempt to extract the maximum tumor measurement specified in the report, if it exists. We have the model retrieve this statement, and automatically parse the value using regex. We then use this data to filter the possible T classes that the report could match.

We then perform a pass for each of the three categories in TNM, supplementing the prompt with labels, definitions, and five example sentences for all possible classes. For the T category, we also indicate the filtered list of categories mentioned above.

For all of the four passes, we force the model to output a valid JSON using the **lm-format-enforcer** library on GitHub (Gat 2023), which allows grammar-based LLM sampling, along with a mode to force the JSON output to match a provided pydantic scheme. This is effectively the same as OpenAI's JSON mode, but on local models without a noticeable impact on performance.

We also prompted the model to cite the statements within the report relevant to its decision. We were able to train for this, as our report synthesis method decided the ground truth labels for both the categorical class and the statements indicating the class.

## Results

As shown in Table 1, vanilla foundational models demonstrate promising but unreliable performance on the task, even in the case of GPT 4. However, even a moderate degree of fine-tuning (100 samples for 16 epochs) increases Mistral 7B Instruct's performance by over 70%. This is in line with existing literature on LoRA, which has shown that the gap in performance between LoRA fine-tuning and full-parameter fine-tuning is very small on more specialized tasks, while it is larger on more generalized and/or reasoning intensive tasks (Niederfahrenhorst, Hakhamaneshi, and Ahmad 2023).

The model's accuracy is reported in Table 1, both separately across T, N, and M categories and compounded for all categories for a single pass. In other words, the compounded metric measures the model's accuracy at correctly predicting all three of the categories in a single inference pass. It is important to note that the accuracy metric includes samples where the correct label is 'UNKNOWN'. We also report an additional metric that measure the fraction of cases in which the model incorrectly predicted 'UNKNOWN' for a category that had a known label, subtracted from one. This measures the "confidence" of the model, with a value of 1.0 indicating that the model did not report 'UNKNOWN' for any case with a known label.

Furthermore, our results showed that training on only 100 samples could yield similar results and enable physicians to efficiently create training and evaluation data sets for similar tasks at a low cost.

## Discussion

The results presented in this study provide compelling evidence that Low-Rank Adaptation (LoRA) fine-tuning can significantly enhance the performance of small foundational language models on specialized tasks such as TNM staging from unstructured pathology reports. Note that compounded accuracy is particularly useful in demonstrating how even marginal error rates become problematic in real-world cases involving multiple steps. Given our best result reached 96.5% compounded accuracy, and given that the task is deterministic in nature, we believe that perfect accuracy is possible with enough training.

It is worth noting that, after fine-tuning, the T category was consistently the least accurate of the three, while the other two quickly converged towards 100% accuracy. This could be attributed to the large number of classes (14 vs 9 and 5 for N and M respectively), however further investigation is warranted into the model's behavior when making mistakes. One could also take the approach of training the model to more easily decide to classify a report as 'UNKNOWN'. During training, the assigned rates at which a category would be hidden were 5% for all categories, and raising this value could sacrifice a marginal level of confidence for improved performance.

## Future Work

### Optimization

It is likely that better configurations may be found for LoRA fine-tuning that result in improved performance over that reported in this work. Likewise, different multi-pass workflows and prompt engineering techniques may further improve performance. We also envision that additional modifications to the data synthesis process to increase the difficulty/variety in evaluation may lead to better generalization.

### Model Confidence

Additionally, we believe it may be advantageous to intentionally train the model to increase its chances of predicting the 'UNKNOWN' label on reports where the decision may be unclear. From a practical standpoint, we recognize that it is better for the model to indicate a lack of confidence than

to generate an incorrect classification. The practical implication of the 'UNKNOWN' category is that the physician would have to manually decide the classification.

## Software Frameworks

This work also involved the development of front-end and back-end software for deploying these local models and their applications. We found that existing software for self-hosting large language models was somewhat limited, so this was necessary in order to securely host these models on local infrastructure at scale. This work has proven extremely promising, and its application may go beyond the scope of the work described here. It is our intention to open source our work once we feel it is production ready, and we may publish future work on our findings.

## Ethical and Operational Considerations

Being a purely technical investigation, this work did not investigate the operational or ethical aspects of deploying such a model in practice. Such investigations should be performed in the event that one seeks to integrate one of these models into their clinical workflow.

## Conclusion

In this paper, we have presented a comprehensive study on the application of Low-Rank Adaptation (LoRA) fine-tuning of small language models, specifically Mistral's 7B model, for the task of TNM staging in unstructured pathology reports of triple-negative breast cancer cases. Our findings have demonstrated that with fine-tuning, even smaller foundational models can achieve high accuracy in medical NLP tasks, offering a cost-effective and reliable alternative to larger, more resource-intensive models controlled by a third party.

The study has shown that a reliable level of performance is possible with just LoRA fine-tuning of small foundational models. Given even a small amount of training data, accuracy becomes substantially better than even the best available foundational models. We've also shown that given a sufficient amount of data, model performance approaches near-perfect accuracy.

## Acknowledgments

We would like to thank the Louisiana Tumor Registry for providing us with the sample reports that we used to construct our synthesis process. We would also like to thank LSU for providing us with the computational resources necessary to train these models. We would also like to thank Yueh-Ming Wang and Aditya Srivastava for helping with development of our deployment frameworks, as well as communication with the Louisiana Tumor Registry.

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
