# OpenReview forum: "TNM Tumor Classification from Unstructured Breast Cancer Pathology Reports using LoRA Finetuning of Mistral 7B"
_AAAI.org/2024/Spring_Symposium_Series/Clinical_FMs — AAAI 2024 SSS on Clinical FMs_

### Official Review · Reviewer_tM9p · 2024-02-14
**A nice study with impressive-looking results, but difficult to follow**

**Rating:** 5
**Confidence:** 2

**Review:**

This is an interesting demonstration of an application of foundation language models cost-effectively fine-tuned to a clinical task and performing impressively, especially compared to language models without fine-tuning.

However, it is hard to follow for someone like myself with machine learning but not medical expertise. For instance, I don't know what TNM or triple-negative mean, and my lack of familiarity with medical reports make the problem and the description of data preparation, which are important to understand the implications of the results, hard to understand. In the results, it is unclear what the difference is between the sample count indicated in the model column vs the samples column. If the former is the number of samples used for training, then the claim that low sample count is sufficient for training seems unsupported, as the accuracy is substantially higher with greater sample count. Also, if possible, it would be helpful if the results included some model trained only on the cancer data so we can tell how much is gained by starting with a foundation model. Further, what are the practical implications of these results? What would be the impact of deploying this model in particular?

A small discussion of prior work in fine-tuning foundation models for clinical tasks and the gap that this work fills could help contextualize this work and understand its contribution. Much of the last page is speculative, which is, I think, less valuable than further supporting the experimental study - the main contribution of this paper - with more details as mentioned above.

---

### Official Review · Reviewer_JW1i · 2024-02-22
**TNM staging using LoRA finetuning**

**Rating:** 8
**Confidence:** 3

**Review:**

The authors performed LoRA finetuning on top of Mistral 7B model to do TNM staging classification tasks on breast cancer pathology reports. The authors carefully curated a dataset of anoymized reports with labels from subject matter experts. The results look very promising. Only one foundational model was used in the finetuning and evaluation, so it is unclear whether there could be significant result difference among different foundation models with different sizes. It is also unclear how well the model is generalized, e.g. whether the quality and the format of the original reports may affect the final results,  though the authors mentioned it in the future work section.

---

### Official Review · Reviewer_c7Sw · 2024-02-22
**An interesting application of LoRa but with questionable data augmentation/quality and limited evaluation**

**Rating:** 6
**Confidence:** 5

**Review:**

Summary
-- the paper applies LoRa fine-tuning of a Mistral model to perform TNM phenotyping using pathology reports.

Pros
-- they perform data augmentation using 200 real world pathology reports. They synthesize new reports by stripping relevant sentences from existing reports and replacing them with example sentences that were mapped to certain label classes a priori
-- the inclusion of an UNKNOWN label class in cases where the relevant information was missing
-- the use of JSON-enforced output
-- reporting of training time as well as performance
-- the ability of the model to cite the relevant information
-- the use of LoRa here is interesting

Cons
-- While the data augmentation method is creative, the methodology is not described clearly enough and the quality of the resulting data is not examined. For example, "These sentences were then injected into the template reports at randomly selected marker locations" -- does this mean that the data is being pulled from a finite list of sentences in the JSON? If so, this is clearly not sufficiently representative of the diversity of natural clinical language.
-- unclear how the path reports were labeled -- what exactly was being labeled, and what were the qualifications of the labelers?
-- how did you strip the report of all info relevant for TNM using a script? How did you validate the accuracy of this process?
-- unclear how many data examples were generated in total. Also not clear whether or not the resulting dataset was high quality. It sounds like you replaced the parts relevant to TNM with random TNM ratings to augment the dataset. Were the resulting pathology reports realistic? It's not obvious that this procedure would result in realistic pathology reports.
-- Some irrelevant text, eg. there is no Section 5
-- This is an interesting application of LoRa, but it's just one task. A more comprehensive evaluation across several tasks would be much more compelling.
-- "We also attempt to develop a more generalized approach, so that our work can be applied to other NLP tasks within the medical field" -- it's not clear where this was done

---

### Official Review · Reviewer_opkq · 2024-02-23
**Meaningful efforts to apply multiple language models to TNM Tumor classification from unstructured reports**

**Rating:** 7
**Confidence:** 3

**Review:**

The paper explored the application of LoRA fine-tuning of Mistral 7B Instruct to classify TNM staging from unstructured pathology reports for triple negative breast cancers. Several baselines and tuning strategies are compared on a real-world data task to demonstrate the proposed fine-tuning approach could achieve better accuracy using small amount of training data.
- Quality: the paper is technically sound and of good quality. Most claims in the paper are well supported by experiment results.
- Clarity: the paper is well-structured and clear in experiment process and results discussion, despite that the paper lacks clarity on the clinical knowledge of TNM  categories (pls refer to the suggestion #1 below).
- Originality: the paper demonstrates an interesting application of well-established techniques to a specific clinical task. Though fine tuning  is not a novel idea, the application to the specific TNM staging seems to be novel from practical perspective.
- Significance: the paper mentioned potential generalization possibility of the proposed approach other clinical tasks, given the low cost and high reliability with comparison to other large language models.
Cons, suggestions or questions to the authors:
1. It'll be better to include a brief introduction of TNM staging (e.g. T stands for size of tumor, etc.) at the beginning of the paper for non-clinical audience, and also why TNM staging of triple negative breast cancers is challenging and demands the help of ML modeling and application. It will also improve the significance of the work from the clinical application perspective.
2. The paper mentioned manual labeling from subject matter experts. Would you pls provide more information on 1) how many experts are involved, 2) whether each document was labeled by multiple experts and how the final label is determined (e.g. if there's any disagreement) . More importantly, could you pls comment how to ensure the reliability and robustness of the proposed fine tuning approach towards label noise in the training data?
3. In table 1, different fine-tuning results show increasing accuracy but decreasing in confidence for "UNKNOWN" class. Does it indicate over-fitting problem?